# Targeting JWA for Cancer Therapy: Functions, Mechanisms and Drug Discovery

**DOI:** 10.3390/cancers14194655

**Published:** 2022-09-24

**Authors:** Kun Ding, Xia Liu, Luman Wang, Lu Zou, Xuqian Jiang, Aiping Li, Jianwei Zhou

**Affiliations:** 1Department of Molecular Cell Biology & Toxicology, Center for Global Health, School of Public Health, Nanjing Medical University, Nanjing 211166, China; 2Key Laboratory of Modern Toxicology of Ministry of Education, School of Public Health, Nanjing Medical University, Nanjing 211166, China; 3Jiangsu Key Lab of Cancer Biomarkers, Prevention and Treatment, Collaborative Innovation Center for Cancer Medicine, Nanjing Medical University, Nanjing 211166, China

**Keywords:** JWA, cancer therapy, tumor growth, angiogenesis, metastasis

## Abstract

**Simple Summary:**

JWA has been identified as a potential therapeutic target for several cancers. In this review, we summarize the tumor suppressive functions of the JWA gene and its role in anti-cancer drug development. The focus is on elucidating the key regulatory proteins up and downstream of JWA and their signaling networks. We also discuss current strategies for targeting JWA (JWA peptides, small molecule agonists, and JWA-targeted Pt (IV) prodrugs).

**Abstract:**

Tumor heterogeneity limits the precision treatment of targeted drugs. It is important to find new tumor targets. JWA, also known as ADP ribosylation factor-like GTPase 6 interacting protein 5 (ARL6IP5, GenBank: AF070523, 1998), is a microtubule-associated protein and an environmental response gene. Substantial evidence shows that JWA is low expressed in a variety of malignancies and is correlated with overall survival. As a tumor suppressor, JWA inhibits tumor progression by suppressing multiple oncogenes or activating tumor suppressor genes. Low levels of JWA expression in tumors have been reported to be associated with multiple aspects of cancer progression, including angiogenesis, proliferation, apoptosis, metastasis, and chemotherapy resistance. In this review, we will discuss the structure and biological functions of JWA in tumors, examine the potential therapeutic strategies for targeting JWA and explore the directions for future investigation.

## 1. Introduction

Cancer has become a serious threat to human health and a major public health problem that needs to be addressed urgently. According to the latest statistics from the International Agency for Research on Cancer (IARC), in 2020, there were an estimated 19.3 million new cancer cases and nearly 10 million deaths worldwide [1]. With an aging population and environmental pollution, global cancer incidence and mortality rates have increased significantly [2]. Despite some advances in targeted cancer therapy, cancer remains the second leading cause of death in the United States and most industrialized countries [3]. Currently, surgery, chemotherapy, radiotherapy, immunotherapy and targeted therapy are the main forms of cancer treatment. However, chemotherapy resistance and the associated toxic side effects, the insensitivity of radiotherapy and immunotherapy, and the limited availability of targeted therapies remain challenges in the recurrence and metastasis of tumors, which seriously affect the survival of patients [4,5,6,7,8,9]. Therefore, there is an urgent need to identify new therapeutic targets for oncology and to develop more effective therapeutic agents based on these targets.

The JWA gene is a cytoskeleton-like gene induced by retinoic acid (RA) and associated with the regulation of cell differentiation; it was isolated and cloned by J Zhou in 1998 and is widely expressed in various tissues and organs [10,11]. The JWA gene is biologically conserved from *C. elegans* to human beings and is involved in many fundamental functions in physiology and biochemistry. JWA regulates target genes by modulating relevant signaling networks to reverse the disrupted microenvironment and can play a crucial role in physiological and pathological states such as cell differentiation, neurological diseases and tumors [11,12,13,14,15,16,17]. Genomic stability is critical for cell survival and tumor progression, yet early identification is challenging [18] since cells are constantly exposed to multiple endogenous and exogenous stimuli, resulting in DNA damage and mutations and thus promoting tumorigenesis. The presence of multiple single nucleotide polymorphisms (SNPs) in the JWA gene may affect the response of the JWA expression to environmental stimuli, negatively affecting DNA damage and repair processes, and ultimately triggering tumor initiation [19,20]. Therefore, the presence of JWA may serve as a marker that can identify the risk of early tumor development. In recent years, substantial evidence implicates JWA as a key tumor suppressor gene during cancer progression. Dysregulation of JWA has been found to be involved in tumor progression through multiple mechanisms, including tumor angiogenesis and metastasis, chemotherapy resistance, apoptosis, and involvement in tumor metabolism [21,22,23,24,25]. More importantly, JWA peptides designed with the JWA active domain structure exhibit powerful suppression of tumor growth, metastasis, and angiogenesis in melanoma and gastric cancer (GC) [26,27,28]. JWA agonists screened with the JWA promoter exhibit a potent ability to inhibit tumor cell proliferation and promote tumor cell apoptosis in breast cancer [29,30]. Consequently, JWA has great potential as a therapeutic target and biomarker for cancer.

In this article, we systematically review for the first time the role of JWA in tumors and anticancer strategies targeting JWA, including its structure, localization, expression, functions, and related signaling networks. We also discuss the regulation of JWA peptides and agonists based on tumor signaling networks. Finally, we present new insights and future research priorities for JWA-targeted anticancer strategies.

## 2. The Structure and Functions of JWA

JWA, also known as ADP ribosylation factor-like GTPase 6 interacting protein 5 (ARL6IP5, GenBank: AF070523, 1998), is a microtubule-associated protein and an environmental response gene [11,31,32,33]. The full-length cDNA open reading frame sequence is 564 bp, encoding 188 amino acids [11]. JWA proteins are highly conserved in humans and other mammals; for example, there is a 95% identity between human and mouse JWA proteins. Homologous genes of JWA, such as glutamate transport-associated protein 3-18 (GTRAP3-18), addicsin, and Jena-Muenchen 4 (JM4), have also been extensively studied for their structure and function [34,35,36]. The JWA gene is located on chromosome 3p14 and contains three exons and two introns. The JWA gene promoter sequence contains multiple response elements, such as TPA (12-O-tetradecanoyl-phorbol-13 acetate) response element (TRE), hemin response element (HRE), heat shock response element (HRE), stress response element (SRE) and ATRA response element (ARE) [11]. JWA belongs to the PRA1 domain family, member 3 (PRAF3), which contains a large prenylated Rab acceptor 1(PRA1) domain [37,38]. Protein structure analysis has shown that JWA is a very hydrophobic protein with three or four transmembrane domains, including protein kinase C (PKC) motifs (SDR-SLR, SDR: codons 18th–20th, SLR: codons 138th–140th) in both C-terminal and N-terminal domains (Figure 1A). JWA protein is located in the endoplasmic reticulum (Figure 1B). Furthermore, JWA binds to a variety of proteins (Figure 1C). The NH_2_ terminus of JWA is important for the binding between JWA and the receptor activator of NF-kB ligand (RANKL), thus inhibiting osteoclastogenesis [39]. The region between 103rd–117th aa of JWA protein is required for homodimer and heterodimer formation with ADP-ribosylation factor-like 6 interacting protein 1 (ARL6IP1) [40]. The region between 145th–188th aa of JWA is required for the formation of the addicsin-TR1 heterocomplex, involved in endoplasmic reticulum stress [41]. GTRAP3-18 was found to regulate the neuronal glutamate transporter excitatory amino acid carrier 1 (EAAC1) through direct protein-protein interactions [42,43]. In addition, JWA proteins also had post-translational modifications (Figure 1D). The PKC phosphorylation motif of JWA is responsible for the activation of the MEK-ERK pathway and cell migration [44,45]; in gastric cancer, JWA lysine 158 is a necessary site for RING Finger Protein 185 (RNF185) ubiquitination and degradation of JWA [46]. This suggests that JWA regulates self-protein stability mainly through protein phosphorylation and ubiquitination of its intracellular signaling molecules [32,37,45].

Functionally, JWA is involved in a wide range of biological processes, including proliferation, differentiation, apoptosis, migration, angiogenesis, DNA damage repair, and drug resistance [22,44,51,52,53,54,55,56,57]. When environmental physicochemical factors (cold and heat stimuli) act on cells, the JWA gene responds rapidly and increases its expression to inhibit oxidative stress and repair DNA damage [12,13,57,58]. In addition, JWA also plays an important role in neuroprotection and osteoblast development [39,42]. Due to its multiple biological functions, JWA is closely associated with a variety of diseases, including neurodegenerative diseases and cancer [14,15,21,59,60]. JWA is a novel tumor suppressor gene, which can inhibit the growth and metastasis of malignant tumors and reverse drug resistance by regulating different signaling networks. This review will focus on the role of JWA in cancer signaling networks, and discuss relevant anticancer strategies based on JWA.

## 3. The Functions of JWA as a Tumor Suppressor Gene

Current research shows that JWA, as a tumor suppressor gene, is mainly closely related to tumorigenesis and progression, including tumorigenesis (genetic susceptibility), tumor progression, apoptosis, angiogenesis, chemotherapy resistance, tumor metabolism, and autophagy (Figure 2). Therefore, a full understanding of the role of JWA in tumor progression will be beneficial for cancer diagnosis and treatment.

### 3.1. JWA Gene Polymorphisms Increase the Risk of Cancer Occurrence

Single nucleotide polymorphisms (SNP) are closely related to tumor susceptibility. JWA gene polymorphisms are associated with the risk of leukemia, esophageal cancer, gastric cancer, and bladder cancer in the Chinese population [19,20,60,61]. In a case-control study of 155 pairs of bladder cancer, multivariate regression analysis revealed that the JWA promoter polymorphism −76G > C was an independent risk factor of bladder cancer [20]. In addition, in another case-control study of 215 bladder cancer patients and 250 control patients, JWA −76G > C, 454C > A and 723T > G were observed to be associated with a significantly increased risk of bladder cancer [61]. In a hospital-based case-control study of leukemia in a South China population, SNP experiments were performed on 202 leukemia patients and 289 controls. An increased risk of leukemia was observed when JWA 454C > A and −76G > C [60]. In an SNP case-control study of gastric cancer and esophageal squamous cell carcinoma in a Chinese population (gastric cancer *n* = 413, esophageal squamous cell carcinoma *n* = 250, cancer-free controls *n* = 814, JWA −76G > C and 723T > G were associated with the risk of gastric cancer and esophageal squamous cell carcinoma [19]. In summary, JWA −76G >C may increase the risk of leukemia, esophageal cancer, gastric cancer and bladder cancer by reducing the transcriptional activity of JWA (Figure 3A). This locus may further help predict and identify the risk of tumor occurrence.

### 3.2. The Role of JWA in Angiogenesis

Angiogenesis is a critical step in cancer development, which is a hallmark of solid tumors and promotes tumor recurrence and metastasis [62,63,64,65]. Previous studies have shown that JWA inhibits angiogenesis in gastric cancer and melanoma [17,21,56]. In GC tissues, higher MVD (CD31+) expression was negatively associated with JWA expression but positively associated with matrix metalloproteinases (MMP2) expression. MMP2 is involved in solid tumor vessel formation and shapes the tumor microenvironment to promote tumor progression [66,67]. Moreover, high expression of JWA inhibits angiogenesis in HUVECs in vitro, and also inhibits neovessel formation in gastric cancer-bearing nude mice and chick embryos in vivo [17]. Mechanistically, JWA promotes the degradation of specificity protein 1 (SP1) through the ubiquitin-proteasome pathway and then inhibits the expression of its downstream pro-angiogenic factor MMP2, ultimately inhibiting the angiogenesis of gastric cancer [17]. Similarly, JWA can inhibit the angiogenesis of melanoma cells both in vivo and in vitro. Due to the heterogeneity of different tumors, JWA mainly inhibits the formation of blood vessels in melanoma by down-regulating the integrin-linked kinase (ILK) signaling pathway through suppressing intergrin αvβ3 and transcription factor SP1 expression (Figure 3B). The ILK signaling pathway can activate the NF-κB/IL6/STAT3/VEGF angiogenesis signaling pathway [21,56]. Overexpression of JWA significantly inhibits the expression of interleukin 6 (IL6) and vascular endothelial-derived growth factor (VEGF), but this inhibition disappears when ILK is also overexpressed. In addition, the combined expression of JWA and ILK could better predict the prognosis of melanoma, and the combined expression of JWA and MMP2 in gastric cancer could also better predict the prognosis of gastric cancer [17,21,56]. In summary, JWA could be used both as a biomarker and an anti-vascular inhibitor for gastric cancer and melanoma [68]. However, whether JWA also exerts anti-angiogenic effects in other tumors, thereby inhibiting tumor progression, still deserves further investigation.

### 3.3. The Role of JWA in the Migration, Invasion, and Metastasis of Cancer

Metastasis is the leading cause of cancer-related death and is a complex process [69,70,71,72,73]. Structurally, as a microtubule-binding protein, JWA regulates migration through the depolymerization and reorganization of microfilaments through the MAPK pathways [32]. Previous studies have demonstrated that JWA can inhibit the migration, invasion, and metastasis of various tumor cells, including melanoma, breast cancer, liver cancer, gastric cancer, and esophageal cancer [16,46,51,74,75]. Lymph node metastasis is a key marker of tumor cell dissemination and a predictor of decreased survival in cancer patients [76,77]. In addition, the low expression of JWA increases the probability of tumor lymph node metastasis and distant metastasis in melanoma, esophageal cancer, liver cancer, and gastric cancer tissues [17,56,75,78]. JWA inhibits the metastasis of melanoma and gastric cancer cells by inhibiting the expression of the transcription factor SP1 and intergrin αvβ3 through suppressing the ILK and MMP2 signaling pathways [16,56]. In HER2-positive gastric cancer cells, JWA inhibits cell migration and actin cytoskeleton reorganization by inhibiting human epidermal growth factor receptor 2 (HER2) expression and its downstream PI3K/AKT signaling pathway [79]. In breast cancer cells, JWA inhibits cell migration and invasion through degradation of C-X-C chemokine receptor 4 (CXCR4) and subsequent inhibition of downstream P-AKT expression [74]. In addition, low expression of JWA inhibited cell invasion by activating focal adhesion kinase (FAK) expression, and the combined expression of JWA and FAK molecules could better predict tumor prognosis in hepatocellular cancer (HCC) [75,80].

In in vivo experiments, knockdown of JWA promotes lung metastasis of melanoma in a tail vein injection model of B16F10 melanoma cells; in addition, knockdown of JWA promotes lymph node metastasis of melanoma cells in a tail vein injection lung metastasis model of human-derived melanoma cells (A375 cells) (lymph node metastasis occurred in 40% of mice in the knockdown JWA group; and none in the control group) [16]. Furthermore, Qiu et al. found that RNF185 enhances lung metastasis by decreasing JWA protein expression levels in a tail vein injection lung metastasis model of gastric cancer (BGC823 cells) [46]. In a hepatocellular carcinoma in situ injection metastasis model, Wu et al. found that lower JWA expression significantly increases hepatic in situ injection lung metastasis [75]. These findings further indicate that JWA markedly reduces lung metastasis as well as lymph node metastasis of a variety of tumors.

Epithelial-mesenchymal transition (EMT) is a dynamic process of cellular transition in which cells lose their epithelial characteristics and acquire mesenchymal characteristics [81]. EMT is associated with a variety of tumor functions, including tumor malignant progression, tumor cell migration, intravascular infiltration, and metastasis [82,83]. Vimentin and β-catenin protein expressions were up-regulated in JWA^−/−^ MEF cells, whereas E-cadherin protein expression levels were down-regulated. JWA may mediate cell migration by regulating the expression of epithelial-mesenchymal transition-related proteins [84]. Therefore, JWA suppresses metastasis involving multiple signaling pathways, including inhibition of receptor-dependent (HER2 and CXCR4) PI3K/AKT activation, regulation of microfilament depolymerization and reorganization, and inhibition of the EMT process (Figure 3C). However, the specific dynamic regulation process of JWA on EMT still needs to be further studied. Taken together, these findings suggest that JWA inhibits metastasis through multiple mechanisms in tumors.

### 3.4. The Role of JWA in Chemotherapy Resistance

Cisplatin-based chemotherapy is currently the main treatment for solid tumors, and drug resistance remains a major obstacle to chemotherapy due to complex pathological features and genomic alterations [85,86]. Compared with parental cells, the expression level of JWA in primary and secondary cisplatin-resistant gastric cancer cells decreases significantly, and JWA enhances cisplatin-induced cell death by regulating DNA damage-induced apoptosis. Mechanistically, JWA promotes X-ray repair cross-complementing group 1(XRCC1) ubiquitination degradation by inhibiting casein-kinase 2 (CK2)-mediated phosphorylation of XRCC1 at 518S/519T/523T, thereby inhibiting XRCC1 overactivation and superior DNA repair ability, and promoting cisplatin-induced apoptosis of drug-resistant cells [22]. Another study found that in cisplatin-resistant GC cells, JWA inhibits tumor necrosis factor-related apoptosis-inducing ligand (TRAIL) by negatively regulating death receptor 4 (DR4) expression. Mechanistically, JWA promotes ubiquitination and degradation of DR4 by upregulating ubiquitin ligase membrane-associated Ring-CH-8 (MARCH8) [23]. Both P-glycoprotein (P-GP) and multidrug resistance-associated protein (MRP) can release energy to pump chemotherapeutic drugs out of the cell through an ATP-dependent mechanism and reduce the concentration of intracellular drugs, thereby reducing the damage of drugs to cells [87]. JWA is found to regulate the expression of P-GP in drug-resistant cells and affect its transport function [88,89]. In addition, JWA alone or in combination with related molecular markers (XRCC1, p53, MDM2, FAK) can predict the prognosis of tumor patients and the sensitivity to chemotherapy drugs [80,90,91,92,93,94]. In summary, in cisplatin-resistant gastric cancer cells, JWA is involved in reversing the chemoresistance process by blocking the XRCC1-dependent DNA repair pathway and promoting the DR4 receptor-dependent apoptosis pathway (Figure 3D).

### 3.5. The Role of JWA in Tumor Metabolism

Metabolic reprogramming of tumor cells is one of the important hallmarks of cancer cell growth and chemoresistance [95]. The imbalance of glucose metabolism, fatty acid synthesis, and glutamine breakdown is closely related to the malignant biological behaviors of cancer cells, such as proliferation, drug resistance, invasion, and metastasis [96,97,98,99]. In addition, multiple tumor-related signaling pathways are involved in the regulation of tumor metabolism [100,101]. High expression of JWA could increase aerobic respiration and adenosine triphosphate (ATP) production, suppress glycolysis, and thus inhibit pancreatic cancer cell metastasis. In addition, JWA can modulate mitochondrial complex III (UQCRC2) to inhibit metastasis. Mechanistically, JWA regulates glycolysis and mitochondrial oxidative phosphorylation through the AMPK (AMP-activated protein kinase)/FOXO3 (Forkhead box O3) pathway to participate in energy metabolism and inhibit tumor metastasis [25,102] (Figure 3E). These findings suggest that JWA may be involved in the metabolic reprogramming of tumor cells to inhibit tumor progression.

### 3.6. The Role of JWA in Cell Proliferation and Apoptosis

Cell proliferation and apoptosis are key features of tumor development [103]. The abnormality of cell cycle progression is one of the basic mechanisms of tumorigenesis, and the regulators of the cell cycle can be a reasonable target for anticancer therapy [104]. JWA promotes arsenic trioxide (As_2_O_3_) and etoposide (VP16)-induced apoptosis and growth inhibition in the tumor cells (HeLa, MCF-7, and JAR) [105]. Mechanistically, JWA triggers apoptosis by regulating tubulin polymerization through the MAPK signaling pathway [105]. JWA also regulates tumor cell apoptosis through reactive oxygen species and mitochondria-related pathways [106]. Likewise, JWA can interact with Bcl-xL/BCL2, and overexpression of JWA can lead to the translocation of BAX to mitochondria and induce tumor cell death [48]. However, whether JWA can also regulate cell death by regulating the stability of Bcl2 deserves further study. Studies have found that JWA could also regulate a variety of cyclins. JWA inhibits the expression of Cyclin-dependent kinase 12 (CDK12), which participates in the transition from G1 to S phase, thus suppressing the proliferation of trastuzumab-resistant breast cancer cells, and promoting their apoptosis [24]. In summary, JWA can regulate the proliferation and apoptosis of tumor cells through a series of signaling networks such as reactive oxygen species, the mitochondrial apoptosis pathway, and cyclin-related regulations (Figure 3F). In the future, JWA agonists may be selectively used in combination with inhibitors of various cyclins for clinical translation.

### 3.7. The Role of JWA in DNA Damage Repair

The DNA damage response is involved in many signaling events, including the regulation of the cell cycle and DNA replication, and is critical for genomic instability as it affects tumorigenesis [107]. DNA damage-inducing therapy is of great value for cancer treatment and functions by directly or indirectly forming DNA damage and subsequently inhibiting cell proliferation. Among the most important cellular responses to treatment-induced DNA damage is the DNA damage response (DDR), a protein network that directs DNA damage repair and induces cancer eradication mechanisms such as apoptosis [108,109]. Wang SY et al. found that JWA could be translocated into the nucleus by the carrier protein XRCC1 after oxidative stress injury and co-localized with XRCC1 foci. In addition, it has been determined that JWA regulates the transcription of nuclear factor E2 promoter binding factor 1 (E2F1) through the MAPK signaling pathway to regulate XRCC1 expression. JWA can be used as a novel regulator of XRCC1 in the BER protein complex to promote DNA SSB repair [110]. In addition, in the process of malignant transformation of MEF cells using JWA knockdown, JWA knockdown causes the upregulation of poly (ADP-ribose) polymerase 1 (PARP1) repair protein and changes in EMT-related proteins, indicating that JWA plays a similar function as a tumor suppressor protein in the process of tumorigenesis [84]. In addition, deletion of JWA in mice aggravates 7,12-dimethylbenz(a)anthracene (DMBA)-induced DNA damage, resulting in more severe genomic instability [111]. These findings suggest a regulatory role for JWA in DNA damage repair and genomic stability. Although there are many synthetic lethal drugs (such as PARP inhibitors) that promote genomic instability [112], whether JWA can similarly disrupt tumor cell repair pathways for antitumor therapy remains an important topic for future research.

### 3.8. The Role of JWA in Autophagy

Dysregulation of autophagy is a double-edged sword during tumor development, where autophagy inhibits cancer initiation but promotes cancer progression. After tumor formation, autophagy activation mediates tumor resistance. Various genes, RNA molecules, proteins, and certain drugs exert antitumor effects by inhibiting autophagy-mediated drug resistance [113,114]. Silencing JWA expression can reduce the expression level of light chain 3 (LC3-Ⅰ/Ⅱ) in TE1 cells after cisplatin treatment, and JWA may affect the sensitivity of esophageal cancer cells to cisplatin by regulating autophagy [115]. These findings suggest that JWA may affect the sensitivity of tumor cells to cisplatin by regulating autophagy, and the specific mechanism needs further investigation.

### 3.9. JWA Exists as a Valid Biomarker

Clinically, most tumors are initially diagnosed as advanced tumors, resulting in a low five-year survival rate. The lack of effective early diagnostic markers limits the early identification of tumors. Therefore, it is particularly important to discover specific biomarkers with high sensitivity and specificity. Numerous studies have shown that JWA is a promising biomarker in a variety of tumors, including gastric cancer, melanoma, and liver cancer [17,56,75]. Chen et al. found a significant negative correlation between JWA expression levels and malignant phenotypes of gastric cancer, including TNM stage, lymph node metastasis, distant metastasis and intratumoral CD31 expression level in a GC training cohort and validation cohort [17]. In addition, overall survival is significantly longer in patients with high intratumoral JWA expression [17]. JWA alone and in combination with XRCC1, FAK, MMP2, MMD2 can predict the prognosis of gastric cancer patients [17,92,93,94]. More importantly, JWA alone and in combination with XRCC1 or in combination with FAK can also be used as a prognostic marker for the outcome of chemotherapy in patients with resectable gastric cancer [80,90]. JWA is lowly expressed in malignant melanoma tissues compared with normal skin [56]. In a melanoma cohort, Lu et al. found that low JWA expression is associated with poor overall survival and five-year survival in a multivariable regression analysis [56]. JWA combined with ILK or ING4 can better predict prognosis in melanoma patients compared with indicators alone [21,56]. In an HCC cohort, Wu et al. found that low expression of JWA is associated with poor clinical case characteristics of HCC, such as tumor size, vascular invasion and TNM stage [75]. Multifactorial regression analysis showed that JWA can be used as an independent prognostic biomarker for overall survival and recurrence-free survival in HCC patients [75]. Nevertheless, large clinical studies still need to be conducted to identify JWA as a prognostic biomarker for cancer patients.

## 4. Anticancer Strategies Targeting JWA

Chemotherapeutic drugs have long been used to kill tumor cells by promoting apoptosis and inhibiting proliferation; however, chemotherapy is unable to distinguish between tumor cells and normal cells, leading to serious toxic side effects in the body. Compared with traditional chemotherapy drugs, targeted drugs can specifically target cancer cells without obviously affecting normal cells, and have high efficiency and low toxicity [116]. Targeted drugs can be broadly classified into two categories: small molecules and large molecules (e.g., monoclonal antibodies, peptides, antibody-drug couples and nucleic acids) [117,118].

### 4.1. JWA Peptide—JP1 and JP3

According to the potential functioning fragments in the coding region of the JWA gene identified in mechanistic evidence, several JWA functional mimic polypeptide fragments have been designed with regular modifications in both ends of the polypeptide fragments to delay their rapid degradation. The characteristics of these polypeptides are modified with the phosphate groups for serine (S), threonine (T) or tyrosine (Y) and designed as small kinase molecules. The subcutaneous tumor-bearing mouse models with the A375 cell line are firstly used for in vivo screening of its anticancer activities by intra-tumoral injection. PJP1 contains seven amino acids from the JWA coding region and shows the best anti-proliferative effect among those candidate fragments. Based on the data of the intra-tumoral injection mouse model, the PJP1 is further modified to form an intergrin targeted peptide (named JP1) by the amino acid triplet Arg-Gly-ASP (RGD) [119,120]. Subsequently, JP1 has shown precise targeting and anticancer activities in subcutaneous tumor-bearing mice by intraperitoneal injection (Figure 4A). In addition, the combination of JP1 and dacarbazine show synergistic inhibition of the tumor-bearing growth of melanoma cells and alleviate dacarbazine-induced liver injury in mice. Furthermore, in a mouse tail vein injection model of the passive metastasis of melanoma, JP1 intraperitoneal injection significantly inhibits lung metastasis in mice, and it prolongs the survival of lung metastasis in melanoma mice. To simulate the development of clinical cancer metastasis, an active metastasis model of lung metastasis with B16F10 was constructed. JP1 was found to significantly inhibit active metastasis when the drug was continued after tumor removal [26]. The results from multiple animal models show that JP1 has a potent anti-metastatic effect and has some clinical application prospects.

Mechanistically, JP1 targets and enters melanoma cells with high expression of integrin αvβ3 through RGD, then interacts with MEK1/2 and further activates the E3 ubiquitinase NEDD4L (neural precursor cell expressed developmentally downregulated 4-like), accelerates the degradation of SP1 via its K685, and finally exerts a transcriptional inhibitory effect on integrin αvβ3 in targeted A375 cells [26]. Integrins are cell adhesion and signaling proteins that play an important role in tumor cell stemness, epithelial cell plasticity, metastasis, and therapeutic resistance [121]. Inhibitors of integrins are currently used in the treatment of cardiovascular disease and inflammatory bowel disease, however, there are significant challenges in cancer [122]. JP1 is an effective peptide for integrin inhibitors, with potential translational applications. Furthermore, in safety testing, no significant changes in body weight are observed in mice after JP1 intraperitoneal intervention, In fact, the mice treated with JP1 are usually show obvious improvements in the indicators of liver function, kidney function and myocardial kinases, suggesting that JP1 has no significant toxicity in the mouse model. No significant adverse effects were observed in healthy crab-eating monkeys injected intravenously with JP1 at a high dose (150 mg/kg) for 14 days (IV, qd), suggesting the biosafety of JP1 [26].

Novel peptides containing histidine-tryptophan-phenylalanine (HWGF) motif specifically recognize MMP2 [123,124]. JWA has shown obvious inhibition of angiogenesis via downregulation of the expression of MMP2 in gastric cancer cells [17]. PJP3 peptide shows an effective tumor suppressive role in subcutaneous tumor-bearing mouse models of gastric cancer cells (Figure 4A); HWGF linked PJP3 (named as JP3) further enhanced its MMP targeting properties. Mechanistically, as an MMP2 targeting peptide, JP3 inhibits angiogenesis through TRIM25 (Tripartite motif containing 25)/SP1/MMP2 signaling, and plays a therapeutic role in gastric cancer [27]. In addition, JP3 competitively inhibits the binding of XRCC1 and CK2 in cisplatin-resistant gastric cancer cells, and JP3 in combination with cisplatin synergistically promotes DNA damage and apoptosis gastric cancer. At the same time, JP3 also protects normal cells from cisplatin-induced DNA damage and apoptosis through the ERK/Nrf2 signaling pathway [28]. Therefore, both JP1 and JP3 may be used in combination with other chemotherapeutic drugs in cancer cells, which not only has a synergistic anti-cancer effect, but also reduces damage to normal organs through anti-oxidative stress signal pathways, which can address two objectives in cancer treatment.

### 4.2. Small Molecular JWA Agonists

Small molecule targeted drugs have advantages compared with large molecule drugs in terms of pharmacokinetic (PK) properties, cost, patient compliance, drug storage, and transport [125,126]. Many studies have shown the deficiency or down regulation of JWA expression in cancer tissues compared with the adjacent normal tissues. Therefore, the agonists screening of the JWA gene at a transcriptional level may be an important way to find JWA gene-based anti-cancer compound candidates. A reporter gene plasmid for the JWA promoter fragment has been successfully constructed and the high throughput assays have been completed in HBE cells. Obviously, this reporter gene plasmid design philosophy is based on maintaining the homeostasis in normal cells and not cancer cells. Both JWA agonist compounds JAC1 and JAC4 are finally selected from multiple candidates based on their lipid-water partition coefficient, water solubility, molecular weight, and activation intensity in HBE cells. Ren et al. recently reported that in HER2-positive breast cancer cells, JAC1 inhibits cell proliferation through the JWA/P38/SMURF1/HER2 signaling pathway [29]. The triple-negative breast cancer (TNBC) is the most aggressive of breast cancer cells. Zhai et al. found that JAC1 effectively suppresses TNBC proliferation and induction of apoptosis through JWA/P38 signaling [30]. The mechanism of JWA activation by JAC1 is identified by specifically binding of JAC1 to Zinc-finger Yin Yang (YY1), which promotes ubiquitinated degradation of YY1, thereby relieving the transcriptional repression on the JWA gene by YY1 and increasing the transcriptional activation of JWA [30]. YY1 regulates genes related to the cell cycle, cell death, and tumor metabolism [127,128,129]; YY1 is highly expressed in many cancers [130]. Accordingly, JAC1 may exert anti-cancer effects in YY1-overexpressed malignant tumors (Figure 4B).

The main side effect of radiation therapy for abdominal malignancy patients is radiotherapy-induced injuries to both the intestinal epithelium and bone marrow hematopoietic function, for which there is no specific clinical treatment [131,132]. Therefore, it is an obvious unmet clinical need to find target drugs for radiation enteritis and bone marrow injury. Zhou et al. recently reported that the JWA agonist JAC4 attenuates intestinal mucosal damage induced by abdominal radiation in mice [133]. More importantly, JAC4 significantly increases the survival rate of mice after exposure to a lethal dose of X-rays. In addition, JAC4 reduces radiation-induced oxidative stress damage and inflammatory responses. In intestinal crypt epithelial cells, JAC4 attenuates radiation-induced ROS production and apoptosis (Figure 4B). In mice with intestinal epithelial knockout JWA, the protective effects of JAC4 against radiation-induced intestinal injury is substantially weakened, suggesting that the protective roles of JAC4 on X-ray irradiation induced intestine epithelium injury depend upon JWA expression [133]. Considering the anti-cancer effect of JWA and its specific function to prevent radiation damage, JAC4 may be considered as a candidate drug for radiotherapy of abdominal malignancies.

The preventive and therapeutic role of dietary therapy of tumors has received significant attention in recent years. Green tea has been reported to exert inhibitory effects in a variety of tumors [134]. Most of the chemo-preventive effects of green tea on cancer have been attributed to polyphenolic compounds, of which (-)-epigallocatechin-3-gallate (EGCG) is the most important [135]. Currently, EGCG has been shown to induce apoptosis and cycle arrest to inhibit tumor cell proliferation and metastasis by inhibiting cancer cell angiogenesis [136,137]. A study by Yuan Li et al. found that EGCG suppresses lung cancer progression by activating JWA expression as well as inhibiting topoisomerase IIα expression in lung cancer cells and in a tumor-bearing model [138]. Mechanistically, the amino acid region 90-188 of JWA is necessary for degradation of topoisomerase IIα. In addition, JWA and topoisomerase IIα can synergistically inhibit lung cancer cell migration and invasion. Therefore, EGCG can exist as a natural compound to activate JWA expression.

### 4.3. Emerging JWA-Targeted Pt (IV) Prodrugs Conjugated with CX-4945

DNA-targeted anti-cancer chemotherapeutic agents (platinum complexes) are currently the most effective chemotherapeutic agents in clinical practice [139]. They work primarily by initiating DNA damage to induce apoptosis using the DNA repair pathways that exist in the cells themselves, which however, may lead to chemotherapy drug resistance [140]. Therefore, the main approach to reversing chemotherapy resistance is to use inhibitors of the DNA repair pathway in combination with platinum-based drugs [141]. Over the years, various approaches have been used to try to increase the selectivity of prodrugs for cancer cells to overcome drug resistance [142]. Guo et al. designed and synthesized two Pt (IV) prodrugs: Cx-platin-Cl and Cx-DN604-Cl [143]. These Pt (IV) prodrugs showed strong toxic effects in tumor cells compared with cisplatin both in vivo and in vitro. In addition, Cx-platin-Cl treatment enhanced T cell infiltration in xenograft mouse tumors, and mechanistically, Pt (IV) prodrugs inhibited the growth of cisplatin-resistant ovarian cancer via the JWA-XRCC1 pathway [143,144]. It has been shown that Pt (IV) prodrugs targeting the JWA-mediated DNA damage repair pathway and immunosuppression can reverse cisplatin resistance. Thus, emerging JWA-targeted Pt (IV) prodrugs can be used as potential immunotherapeutic agents for tumor resistance (Figure 4C).

## 5. Conclusions and Prospects

This review summarizes and highlights research findings on the interaction of JWA with hallmarks of cancer, identifying it as a potential therapeutic activating target. JWA inhibits many processes that can drive tumor progression, including inhibition of invasion and angiogenesis, reversal of chemotherapy resistance, promotion of tumor cell apoptosis, and involvement in tumor metabolic reprogramming processes. Based on the association of JWA with tumor characteristics, peptides, small molecules, and prodrugs that target JWA have shown powerful therapeutic effects.

Since JWA contains protein kinase C (PKC) motifs in both the C-terminal and amino acid-terminal domains, it can regulate tumor growth through various signaling pathways, such as the AMPK signaling pathway, MEK/ERK signaling pathway, p38 signaling pathway, and PI3K/AKT signaling pathway. However, the exact modification sites of JWA need to be further investigated. In addition, peptides and compounds designed to target JWA-related proteins MMP2, integrins, and transcription factor YY1 have shown powerful therapeutic effects in tumors. However, due to the heterogeneity of tumors, treatment against a single target usually results in limited improvements in patients, so the combination of JWA peptides or compounds and other anticancer drugs, may be of more benefit since JWA peptides and agonists may also reduce the toxicities of anticancer drugs to normal organs of patients. Furthermore, JWA has the effect of reversing chemotherapy drug resistance and repairing chemotherapy drug-induced liver and kidney injury, and the combination of chemotherapy drugs and JWA peptides can be further used for patients with a later stage of tumor development, and can provide a basis for a new strategy to effectively reverse drug resistance. Of course, the roles of JWA protein-based candidate peptides and agonists and treatment strategies need to be further investigation as radiotherapy insensitivity and chemoresistance are major barriers to effective therapy. More importantly, current studies have mainly focused on key molecular pathways downstream of JWA, with upstream pathways largely comprising transcriptional regulation (ILK, YY1) and post-translational modifications (RNF185), the presence of epigenetic regulation or other post-translational modifications of JWA needs to be further elucidated.

## Figures and Tables

**Figure 1 cancers-14-04655-f001:**
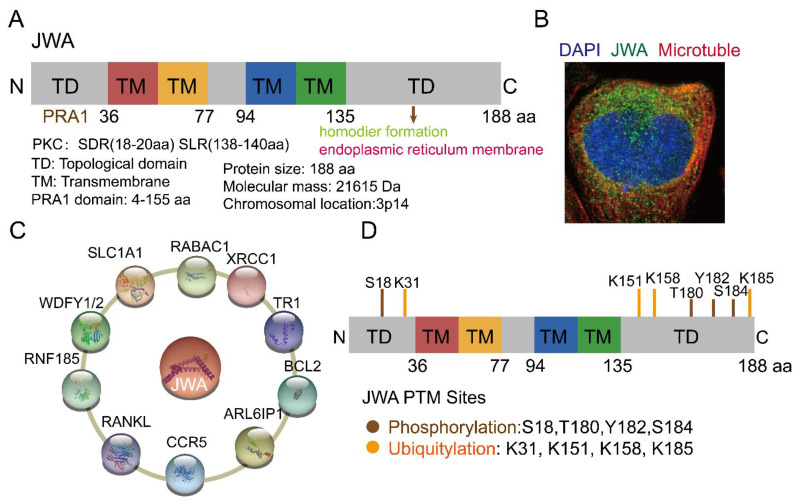
Structure, location and potential modification sites of JWA. (**A**) Domain structure of JWA. JWA consists of four transmembrane (TM) domains, two topological domains (TD) in N terminal (1–36 aa) and C terminal (135–188 aa). JWA also has protein kinase C (PKC) motifs (SDR-SLR) in both C-terminal and N-terminal domains. (**B**) The location of JWA in cells from The Human Protein Atlas (https://www.proteinatlas.org/ENSG00000144746-ARL6IP5/subcellular) (accessed on 2 August 2022)) [47]. The main location of JWA is the endoplasmic reticulum. Confocal imaging shows JWA (green), microtubules (red) and nucleus staining with DAPI (blue). (**C**) Binding protein of JWA. JWA binds to multiple proteins and participates in the regulation of signaling networks [22,35,38,39,40,41,46,48,49]. (**D**) Potential modification sites of JWA from the Phosphosite tool (http://www.phosphosite.org (accessed on 15 August 2022)) [50]. Among these, JWA K158 is required for its ubiquitination and degradation by E3 ubiquitin ligase RNF185. In addition, JWA has phosphorylation sites at the C- and N-terminal domains, which can stabilize its expression.

**Figure 2 cancers-14-04655-f002:**
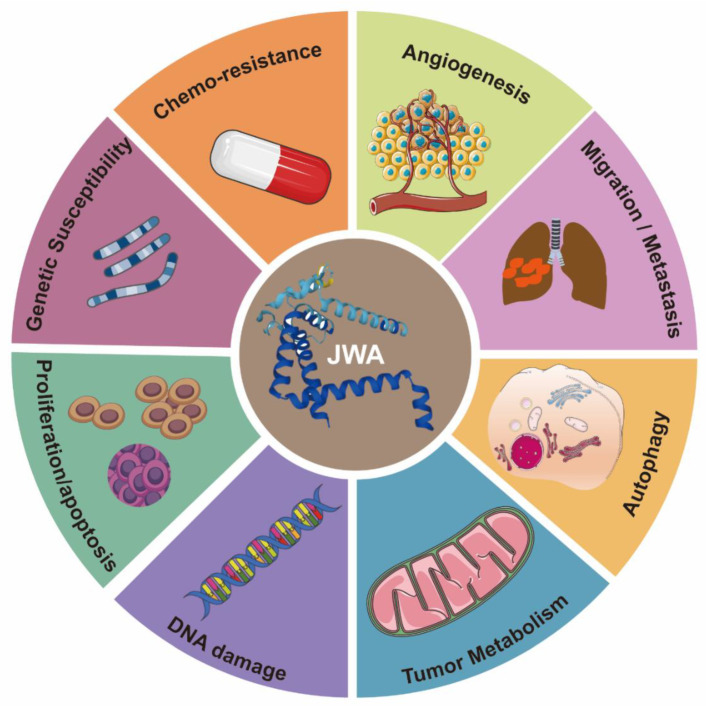
The biological functions of JWA in cancer. JWA inhibits tumor progression by affecting multiple hallmarks of cancer, including angiogenesis, chemo-therapy, genetic susceptibility, proliferation, apoptosis, DNA damage, tumor metabolism, autophagy and tumor metastasis.

**Figure 3 cancers-14-04655-f003:**
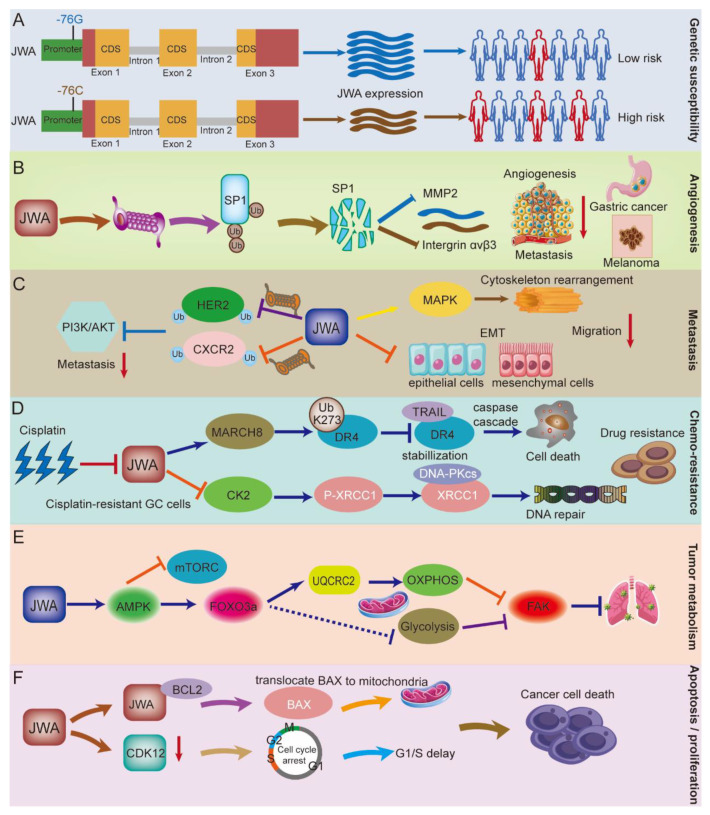
Molecular mechanisms of JWA in cancer. (**A**) Genetic susceptibility: JWA can act as a biomarker and therapeutic target. Mutations in the JWA promoter region (−76G > C) lead to genomic stability and reduce expression of the oncogene JWA, thereby increasing the incidence of cancers (leukaemia, oesophageal cancer, gastric cancer) in the population. (**B**) Angiogenesis: JWA inhibits angiogenesis and subsequent metastasis in gastric cancer and melanoma by the specific mechanism that JWA ubiquitinates and degrades SP1 expression, thus suppressing the transcription of MMP2 and intergrin αvβ3. (**C**) Tumor metastasis: JWA suppresses metastasis involving multiple signaling pathways, including inhibition of receptor-dependent (HER2 and CXCR4) PI3K/AKT activation, regulation of microfilament depolymerization and reorganization, and inhibition of the EMT process. (**D**) Chemotherapy resistance: in cisplatin-resistant gastric cancer cells, JWA is involved in reversing the chemoresistance process by blocking the XRCC1-dependent DNA repair pathway and promoting the DR4 receptor-dependent apoptosis pathway. (**E**) Tumor metabolism: JWA enhances cellular oxidative phosphorylation and inhibits the glycolytic process through the AMPK/FOXO3a signaling pathway, thereby participating in the metabolic reprogramming of cancer cells. (**F**) Apoptosis: JWA promotes cell apoptosis and inhibits cell proliferation in cancer through the mitochondrial apoptosis pathway and cell cycle arrest.

**Figure 4 cancers-14-04655-f004:**
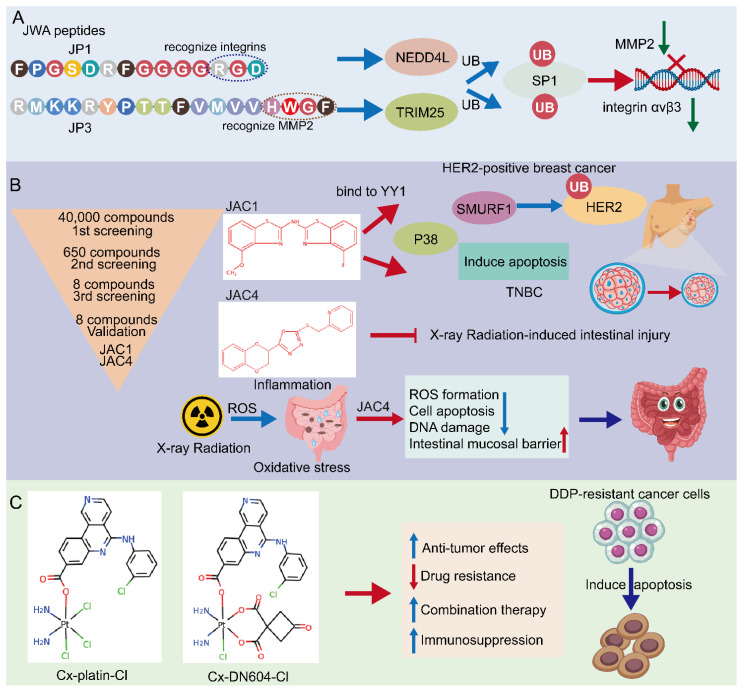
Potential approaches for JWA targeting in cancers. (**A**) Screening of JWA-targeted peptides: mechanistic study of anti-metastatic peptide JP1 and anti-angiogenetic peptide JP3. (**B**) Screening of JWA agonists: JAC1 has pro-apoptotic effect in HER2-positive breast cancer and TNBC. Moreover, JAC4 can attenuate X-ray radiation-induced intestinal epithelium injury by JWA-mediated anti-oxidation/inflammation signaling. (**C**) Emerging JWA-targeted Pt (IV) conjugated with CK2 inhibitor CX-4945: CX-platin-Cl and CX-DN604-Cl can overcome chemo-immune-resistance.

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
