# Peer review of "Targeting JWA for Cancer Therapy: Functions, Mechanisms and Drug Discovery"

_cancers, 2022, doi:10.3390/cancers14194655_

Round 1

Reviewer 1 Report

The review by Ding et al. efficiently summarizes the functions of JWA in cancer. In addition to outlining basic information on JWA, the review sufficiently elucidates its function in cancers, especially in gene polymorphism, angiogenesis, metastasis, drug resistance, metabolism, and cell proliferation. Although I find this review potentially interesting, instances of stretching the meaning of the original data could be observed throughout the manuscript. The in vivo experiments, such as tumorigenesis and metastasis, also need to be carefully explained. Otherwise, the review seems to represent JWA only as negative regulator of tumors. Furthermore, observation studies on human materials in the review are limited. The review article should ideally be based upon present publications, without theological speculation. Although JWA peptides (JP1 and JP3) are interesting research area, their primary functional mechanism seems to differ from endogenous JWA. Hence, if the authors want to emphasize cancer therapy, JWA peptides should be mainly mentioned. 

Apart from these, I have a couple of specific concerns: 

1. The location of the JWA in Figure 1B is unclear. Also, the resources for Figure 1C need to be indicated accurately. Are they drawn from any previous work? All publications or databases need to be properly referenced.

2. In each section, the introductions comprising general explanations of terms, such as angiogenesis and metastasis, might be redundant for the readers of Cancers

Author Response

Reviewer#2

The review by Ding et al. efficiently summarizes the functions of JWA in cancer. In addition to outlining basic information on JWA, the review sufficiently elucidates its function in cancers, especially in gene polymorphism, angiogenesis, metastasis, drug resistance, metabolism, and cell proliferation. Although I find this review potentially interesting, instances of stretching in the meaning of the original data could be observed throughout the manuscript. The in vivo experiments, such as tumorigenesis and metastasis, also need to be carefully explained. Otherwise, the review seems to represent JWA only as the regulator of tumors. Furthermore, observation studies on human materials in the review are limited. The review article should ideally be based upon present publications, without theological speculation. Although JWA peptides (JP1 and JP3) are interesting research area, their primary functional mechanism seems to differ from endogenous JWA. Hence, If the authors want to emphasize cancer therapy, JWA peptides should be mainly mentioned.

  1. The in vivo experiments, such as tumorigenesis and metastasis, also need to be carefully explained.

Response: We do appreciate the reviewer for the comment and suggestions. This suggestion has been well taken. In vivo experiments, knockdown of JWA promotes lung metastasis of melanoma in a tail vein injection model of B16F10 melanoma cells; In addition, knockdown of JWA promotes lymph node metastasis of melanoma cells in a tail vein injection lung metastasis model of human-derived melanoma cells (A375 cells) (lymph node metastasis occurs in 40% of mice in the knockdown JWA group, and none in the control group) [1]. Furthermore, Qiu et al. found that RNF185 enhances lung metastasis by decreasing JWA protein expression levels in a tail vein injection lung metastasis model of gastric cancer (BGC823 cells) [2]. In a hepatocellular carcinoma in situ injection metastasis model, Wu et al. found that lower JWA expression significantly increases hepatic in situ injection lung metastasis [3]. These findings further indicate that JWA markedly reduces lung metastasis as well as lymph node metastasis of a variety of tumors. This part regarding JWA in metastasis has been added in lines 187-197, page 4 of the revised manuscript. Regarding the JWA in tumorigenesis in vivo experiments, to investigate the role and the underlying mechanisms of JWA as a DNA repair protein and the significant role of JWA in skin carcinogenesis, Gong et al. found that JWA deficiency attenuates the development of mouse skin papilloma and inhibits DMBA/TPA induced cell proliferation [4]. Mechanistically, JWA deficiency enhances DNA damage in epidermal cells induced by DMBA, however, suppresses TPA-induced MEK-ERK and transcriptional factor EIK1 activation. This part regarding JWA in tumorigenesis has been added lines 285-288, page 6 of the revised manuscript.

Reference:

  1. Bai, J.; Zhang, J.; Wu, J.; Shen, L.; Zeng, J.; Ding, J.; Wu, Y.; Gong, Z.; Li, A.; Xu, S.; et al. Jwa regulates melanoma metastasis by integrin alphavbeta3 signaling. Oncogene 2010, 29, 1227-1237.
  2. Qiu, D.; Wang, Q.; Wang, Z.; Chen, J.; Yan, D.; Zhou, Y.; Li, A.; Zhang, R.; Wang, S.; Zhou, J. Rnf185 modulates jwa ubiquitination and promotes gastric cancer metastasis. Biochim Biophys Acta Mol Basis Dis 2018, 1864, 1552-1561.
  3. Wu, X.; Chen, H.; Gao, Q.; Bai, J.; Wang, X.; Zhou, J.; Qiu, S.; Xu, Y.; Shi, Y.; Wang, X.; et al. Downregulation of jwa promotes tumor invasion and predicts poor prognosis in human hepatocellular carcinoma. Mol Carcinog 2014, 53, 325-336.
  4. Gong, Z.; Shi, Y.; Zhu, Z.; Li, X.; Ye, Y.; Zhang, J.; Li, A.; Li, G.; Zhou, J. Jwa deficiency suppresses dimethylbenz[a]anthracene-phorbol ester induced skin papillomas via inactivation of mapk pathway in mice. Plos One 2012, 7, e34154.

  1. Furthermore, observation studies on human materials in the review are limited.

Response: We thank the reviewer for the comment and suggestions. This suggestion has been well taken. Numerous studies have shown that JWA is a promising biomarker in a variety of tumors, including gastric cancer, melanoma, and liver cancer [1,2,3]. Chen et al. found a significant negative correlation between JWA expression level and malignant phenotype of gastric cancer, including TNM stage, lymph node metastasis, distant metastasis and intratumoral CD31 expression level in GC training cohort and validation cohort [1]. In addition, overall survival was significantly longer in patients with high intratumoral JWA expression [1]. JWA alone and in combination with XRCC1, FAK, MMP2, MMD2 can predict the prognosis of gastric cancer patients [1,4,5,6]. More importantly, JWA alone and in combination with XRCC1 or in combination with FAK can also be used as a prognostic marker for the outcome of chemotherapy in patients with resectable gastric cancer [7,8]. JWA is lowly expressed in malignant melanoma tissues compared to normal skin [2]. In a melanoma cohort, Lu et al. found that low JWA expression is associated with poor overall survival and five-year survival in multivariable regression analysis [2]. JWA combined with ILK or ING4 can better predict prognosis in melanoma patients compared to indicators alone [2,9]. In an HCC cohort, Wu et al. found that low expression of JWA is associated with poor clinical case characteristics of HCC, such as tumor size, vascular invasion and TNM stage [3]. Multifactorial regression analysis showed that JWA can be used as an independent prognostic biomarker for overall survival and recurrence-free survival in HCC patients [3]. Nevertheless, large clinical studies still need to be conducted to identify JWA as a prognostic biomarker for oncology patients. This part regarding JWA studies on human materials has been added in lines 302-326, page 7 of the revised manuscript.

Reference:

  1. Chen, Y.; Huang, Y.; Huang, Y.; Xia, X.; Zhang, J.; Zhou, Y.; Tan, Y.; He, S.; Qiang, F.; Li, A.; et al. Jwa suppresses tumor angiogenesis via sp1-activated matrix metalloproteinase-2 and its prognostic significance in human gastric cancer. Carcinogenesis 2014, 35, 442-451.
  2. Lu, J.; Tang, Y.; Farshidpour, M.; Cheng, Y.; Zhang, G.; Jafarnejad, S.M.; Yip, A.; Martinka, M.; Dong, Z.; Zhou, J.; et al. Jwa inhibits melanoma angiogenesis by suppressing ilk signaling and is an independent prognostic biomarker for melanoma. Carcinogenesis 2013, 34, 2778-2788.
  3. Wu, X.; Chen, H.; Gao, Q.; Bai, J.; Wang, X.; Zhou, J.; Qiu, S.; Xu, Y.; Shi, Y.; Wang, X.; et al. Downregulation of jwa promotes tumor invasion and predicts poor prognosis in human hepatocellular carcinoma. Mol Carcinog 2014, 53, 325-336.
  4. Wang, W.; Yang, J.; Yu, Y.; Deng, J.; Zhang, H.; Yao, Q.; Fan, Y.; Zhou, Y. Expression of jwa and xrcc1 as prognostic markers for gastric cancer recurrence. Int J Clin Exp Pathol 2020, 13, 3120-3127.
  5. Liu, X.; Wang, S.; Xia, X.; Chen, Y.; Zhou, Y.; Wu, X.; Zhang, J.; He, S.; Tan, Y.; Qiang, F.; et al. Synergistic role between p53 and jwa: prognostic and predictive biomarkers in gastric cancer. Plos One 2012, 7, e52348.
  6. Ye, Y.; Li, X.; Yang, J.; Miao, S.; Wang, S.; Chen, Y.; Xia, X.; Wu, X.; Zhang, J.; Zhou, Y.; et al. Mdm2 is a useful prognostic biomarker for resectable gastric cancer. Cancer Sci 2013, 104, 590-598.
  7. Chen, Y.; Xia, X.; Wang, S.; Wu, X.; Zhang, J.; Zhou, Y.; Tan, Y.; He, S.; Qiang, F.; Li, A.; et al. High fak combined with low jwa expression: clinical prognostic and predictive role for adjuvant fluorouracil-leucovorin-oxaliplatin treatment in resectable gastric cancer patients. J. Gastroenterol. 2013, 48, 1034-1044.
  8. Wang, S.; Wu, X.; Chen, Y.; Zhang, J.; Ding, J.; Zhou, Y.; He, S.; Tan, Y.; Qiang, F.; Bai, J.; et al. Prognostic and predictive role of jwa and xrcc1 expressions in gastric cancer. Clin. Cancer Res. 2012, 18, 2987-2996.
  9. Lu, J.; Tang, Y.; Cheng, Y.; Zhang, G.; Yip, A.; Martinka, M.; Dong, Z.; Zhou, J.; Li, G. Ing4 regulates jwa in angiogenesis and their prognostic value in melanoma patients. Br J Cancer 2013, 109, 2842-2852.

  1. The location of the JWA in Figure 1B is unclear. Also, the resources for Figure 1C need to be indicated accurately. Are they drawn from any previous work? All publications or databases need to be properly referenced.

Response: We do appreciate the reviewer’s comment and interest in this point. We have replaced the obvious images in Figure 1B. Figure 1C is derived from previously published literature (RNF185, RANKL, CCR5, ARL6IP1, BCL2, TR1, XRCC1,RABAC1, SCL1A1) [1-8] as well as database reports (WDFY1/2) [9]. We have inserted literature in Figure 1C legends (line 395, page 9) as requested in the revised manuscript.

Reference:

  1. Qiu, D.; Wang, Q.; Wang, Z.; Chen, J.; Yan, D.; Zhou, Y.; Li, A.; Zhang, R.; Wang, S.; Zhou, J. Rnf185 modulates jwa ubiquitination and promotes gastric cancer metastasis. Biochim Biophys Acta Mol Basis Dis 2018, 1864, 1552-1561.
  2. Wu, Y.; Wang, M.; Peng, Y.; Ding, Y.; Deng, L.; Fu, Q. Overexpression of arl6ip5 in osteoblast regulates rankl subcellualr localization. Biochem Biophys Res Commun 2015, 464, 1275-1281.
  3. Schweneker, M.; Bachmann, A.S.; Moelling, K. Jm4 is a four-transmembrane protein binding to the ccr5 receptor. FEBS Lett 2005, 579, 1751-1758.
  4. Akiduki, S.; Ikemoto, M.J. Modulation of the neural glutamate transporter eaac1 by the addicsin-interacting protein arl6ip1. J. Biol. Chem. 2008, 283, 31323-31332.
  5. Vento, M.T.; Zazzu, V.; Loffreda, A.; Cross, J.R.; Downward, J.; Stoppelli, M.P.; Iaccarino, I. Praf2 is a novel bcl-xl/bcl-2 interacting protein with the ability to modulate survival of cancer cells. Plos One 2010, 5, e15636.
  6. Arano, T.; Fujisaki, S.; Ikemoto, M.J. Identification of tomoregulin-1 as a novel addicsin-associated factor. Neurochem. Int. 2014, 71, 22-35.
  7. Xu, W.; Chen, Q.; Wang, Q.; Sun, Y.; Wang, S.; Li, A.; Xu, S.; Roe, O.D.; Wang, M.; Zhang, R.; et al. Jwa reverses cisplatin resistance via the ck2-xrcc1 pathway in human gastric cancer cells. Cell Death Dis. 2014, 5, e1551.
  8. Koomoa, D.L.; Go, R.C.; Wester, K.; Bachmann, A.S. Expression profile of praf2 in the human brain and enrichment in synaptic vesicles. Neurosci. Lett. 2008, 436, 171-176.
  9. Szklarczyk, D.; Gable, A.L.; Nastou, K.C.; Lyon, D.; Kirsch, R.; Pyysalo, S.; Doncheva, N.T.; Legeay, M.; Fang, T.; Bork, P.; et al. The string database in 2021: customizable protein-protein networks, and functional characterization of user-uploaded gene/measurement sets. Nucleic Acids Res. 2021, 49, D605-D612.

  1. Although JWA peptides (JP1 and JP3) are interesting research area, their primary functional mechanism seems to differ from endogenous JWA. Hence, If the authors want to emphasize cancer therapy, JWA peptides should be mainly mentioned.

Response: We do appreciate the reviewer’s comment and interest in this point. Indeed, JWA polypeptides are selected from the amino acid fragments of JWA, so JP1 and JP3 are different from the JWA protein itself, and belong to the active fragments of the JWA protein, which can partially exert the function of JWA. In addition, in order to increase the content of JWA peptides, we further increase the safety of peptides and the discussion of animal models of peptides. This part has been added in Section 4.1(lines 351-358, pages 7-8; lines 368-372, page 8) in the revised manuscript.

  1. In each section, the introductions comprising general explanations of terms, such as angiogenesis and metastasis, might be redundant for the readers of Cancers.

Response: We thank the reviewer for the comment and suggestions. We have further streamlined the description of the terms of metastasis and angiogenesis in the revised manuscript.

Reviewer 2 Report

This is an interesting review manuscript about the potential of JWA as an anticancer target and for the design of new cancer treatments. The manuscript is well designed. However, it has some shortcomings. Hence, I recommend acceptance after major revision:

Line 88 and Figure 4A: Has the protein protein kinase C motifs or PKC phosphorylation motifs? Please clarify.

Line 101: The authors mention a crucial lysine 158 of JWA important for stability. Is anything known about acetylation of this residue, which might be a hint at a regulation of JWA by HDACs or HATs? If not, this might be a matter for the Prospects section because various HDAC inhibitors were already approved for anticancer therapy.

Section 4.2.: Please also mention the activation of JWA by natural compounds such as EGCG (Li et al., Sci. Rep. 2015, 5, 11099).

Line 426: Please replace ´´platinum´´ by ´´platinum complexes´´.

Figure 4A: Please correct ´´intergrins´´ and ´´intergrin´´.

Figure 4C: The structure of Cx-DN604-Cl is apparently wrong. Please correct.

Abbreviations: The abbreviations should be listed in alphabetical order. Readers might wonder why JWA is missing here.

References 7 and 9: Article ID numbers are missing. Please add.

Reference 11: Authors? Volume? Page numbers?

References 45, 92, 106, 117, 127, 129, 131, and 137: Page numbers or ID numbers?

Author Response

Reviewer#1

This is an interesting review manuscript about the potential of JWA as an anticancer target and for the design of new cancer treatments. The manuscript is well designed. However, it has some shortcomings.

Specific comments:

  1. Line 88 and Figure 4A: Has the protein kinase C motifs or PKC phosphorylation motifs? Please clarify.

Response: We thank the reviewer for the comment. Indeed, JWA has protein kinase C (PKC) motifs (SDR-SLR, SDR: codons 18th-20th, SLR: codons 138th-140th) in both C-terminal and N-terminal domains [1]. These motifs have been clarified in line 88, pages 2 of the revised manuscript.

Reference:

[1] Chen, H.; Bai, J.; Ye, J.; Liu, Z.; Chen, R.; Mao, W.; Li, A.; Zhou, J. Jwa as a functional molecule to regulate cancer cells migration via mapk cascades and f-actin cytoskeleton. Cell. Signal. 2007, 19, 1315-1327.

  1. Line 101: The authors mention a crucial lysine 158 of JWA important for stability. Is anything known about acetylation of this residue, which might be a hint at a regulation of JWA by HDACs or HATs? If not, this might be a matter for the prospects section because HDACs inhibitors were already approved for anticancer therapy.

Response: We do appreciate the reviewers’ comments and interest in this point. The current research on this site of JWA (lysine 158) only stops at ubiquitination, and the stability of JWA mediated by this site is mediated by the E3 ubiquitinase RNF185. The acetylation study of this site in JWA has not been reported in the previous literature, and further research is still needed.

Reference:

  1. Qiu, D.; Wang, Q.; Wang, Z.; Chen, J.; Yan, D.; Zhou, Y.; Li, A.; Zhang, R.; Wang, S.; Zhou, J. Rnf185 modulates wa ubiquitination and promotes gastric cancer metastasis. Biochim Biophys Acta Mol Basis Dis 2018, 1864, 1552-1561.

  1. Section 4.2.: Please also mention the activation of JWA by natural compounds such as EGCG (Li et al., Sci. Rep. 2015, 5, 11099).

Response: This suggestion has been well taken. A study by Yuan Li et al. found that EGCG suppresses lung cancer progression by activating JWA expression as well as inhibiting topoisomerase IIα expression in lung cancer cells and in a tumor-bearing model [1]. Mechanistically, the amino acid region 90-188 of JWA is necessary for degradation of topoisomerase IIα. In addition, JWA and topoisomerase IIα can synergistically inhibit lung cancer cell migration and invasion. Therefore, EGCG can exist as a natural compound to activate JWA expression. The point has been further discussed in lines 468-480, page 13 of the revised manuscript.

Reference:

[1] Li, Y.; Shen, X.; Wang, X.; Li, A.; Wang, P.; Jiang, P.; Zhou, J.; Feng, Q. Egcg regulates the cross-talk between jwa and topoisomerase iiα in non-small-cell lung cancer (nsclc) cells. Sci. Rep.-Uk 2015, 5, 11009.

  1. Line 426: Please replace “platinum” by “platinum complexes”.

Response: We thank the reviewer for pointing out this mistake. It has been replaced in Line 482, page 13 of the revised manuscript.

  1. Figure 4A: Please correct “intergrins” and “intergrin”.

Response: We thank the reviewer for pointing out this mistake. It has been corrected in Figure 4A.

  1. Figure 4C: The structure of Cx-DN604-Cl is apparently wrong. Please correct.

Response: We thank the reviewer for pointing out this mistake. It has been corrected in Figure 4C.

  1. Abbreviations: The abbreviations should be listed in alphabetical order. Readers might wonder why JWA is missing here.

Response: This point has been well taken. We have been listed the abbreviations in alphabetical order and added JWA abbreviation in Line 536, pages 14-15 of the revised manuscript.

  1. References 7 and 9: Article ID numbers are missing. Please add.

Response: We thank the reviewer for pointing out this mistake. It has been added.

  1. Reference 11: Authors? Volume? Page numbers?

Response: We thank the reviewer for pointing out this mistake. It has been added.

10.Reference 45, 92, 106, 117, 127, 129, 131, and 137: Page numbers or ID numbers?

Response: We thank the reviewer for pointing out this mistake. It has been added.

Round 2

Reviewer 1 Report

All my concerns are well addressed. 

Reviewer 2 Report

The revised manuscript can be accepted now.

Only in Figure 4C, there is still a tiny ´´inorganic chemistry´´ mistake the authors can correct during the correction of the proofs (or before): the ammine ligands of the platinum complexes should be described as ´´NH3´´ and not as ´´NH2´´.